# Spatial heterogeneity of menstrual discriminatory practices against Nepalese women: A population-based study using the 2022 Demographic and Health Survey

**Geoffrey Barini**[1]*, **Sharon Amima**[2], **Damaris Mulwa**[3], **Polycarp Mogeni**[4]

**1** Department of Pure and Applied Mathematics, Jomo Kenyatta University of Agriculture and Technology, Nairobi, Kenya, **2** Department of Food Science, Nutrition and Technology, University of Nairobi, Nairobi, Kenya, **3** Department of Statistics and Actuarial Sciences, Jomo Kenyatta University of Agriculture and Technology, Nairobi, Kenya, **4** Kenya Medical Research Institute (KEMRI), Nairobi, Kenya

* barini@jkuat.ac.ke

## Abstract

Menstrual discrimination hampers progress toward Sustainable Development Goals. Examining the spatial heterogeneity of menstrual discriminatory practices may present an opportunity for targeted interventions. Here we evaluate geographical disparities in menstrual-related restrictions and assess their association with socio-economic and demographic factors. We used data from the 2022 Nepal Demographic and Health Survey which included 13,065 women aged 15–49 who reported menstruating within the past year. We explored the spatial heterogeneity of menstrual restriction outcomes using the standard Gaussian kernel density approximation method and the spatial scan statistic. The Poisson regression model with robust standard errors was used to assess the association between the different forms of menstrual restriction and the socio-economic, and demographic factors. Overall, the prevalence of women who reported any form of menstrual restriction was 84.8% and was subject to geographical variations ranging from 79.0% in Bagmati to 95.6% in Sudurpashchim. Religious restrictions were the most prevalent (79.8%) followed by household-level restrictions (39.5%) and then *Chhaupadi* (6.2%). Geographical variations were more prominent for women experiencing Chhaupadi (primary geographical cluster: relative risk = 7.4, p<0.001). Strikingly, women who reside in households led by female household heads were less likely to report experiencing household-level restriction during menstruation (Adjusted prevalence ratio (aPR) = 0.89, [95%CI: 0.84–0.94], p<0.001) whilst those residing in wealthy households were less likely to report experiencing Chhaupadi (aPR = 0.26, [95%CI: 0.17–0.39], p<0.001; among the richest). Our study demonstrated marked geographical micro-variations in menstrual discriminatory practices in Nepal. Policymakers should implement preventive behavioral interventions in the most vulnerable geographic areas to effectively and efficiently reduce the overall prevalence of menstrual discrimination. It is crucial to prioritize the designing and testing of targeted interventions to determine their effectiveness against Chhaupadi in these high-prevalence settings. Additionally, empowering women appears to be a promising strategy for combating menstrual discrimination within the household.

**Data Availability Statement:** All health data used in this analysis are available to download free of charge by registered users from the Demographic

and Health Surveys Program. Registration, instructions on data access are available online at https://dhsprogram.com/data/new-user-registration.cfm. For data see https://dhsprogram.com/data/dataset_admin/download-manager.cfm. For full dataset access instructions see https://dhsprogram.com/data/Using-DataSets-for-Analysis.cfm.

**Funding:** The authors received no specific funding for this work.

**Competing interests:** The authors have declared that no competing interests exist.

## Introduction

Menstrual health is an important public health and human rights challenge, influencing physical and mental well-being, gender equality, and socioeconomic participation [1]. Inadequate menstrual health management contributes to health disparities, educational barriers, and infringes on dignity and equal human rights [2]. Despite the clear evidence, menstrual health and hygiene continue to pose significant public health and social challenges in Low- and Middle-Income Countries (LMIC)[2–5]. Therefore, addressing menstrual health barriers remains essential for achieving the United Nations Sustainable Development Goals (SDGs), particularly those related to health, education, gender equality, and reducing inequalities[6].

Across LMIC, millions of women continue to encounter difficulties in managing their menstrual periods while preserving their dignity. This is predominantly attributed to period poverty [7–12], internalized stigma, shame rooted in myths and misconceptions surrounding menstruation [2, 13–18]. These myths and misconceptions, which regard menstrual blood as a contaminant are deeply rooted in the cultural and religious beliefs [19–25] among some Asian communities leading to discriminatory practices against menstruating women and girls[26–28].

In the South Asian context, Nepal is one of the most affected countries, with over 90% of menstruating girls and women experiencing some form of discriminatory practices during their menstrual periods [29]. These practices are spread across many South Asian countries where menstruation is considered taboo [19, 23, 24, 30–32]], though with varying degrees of intensity. These menstrual discriminatory practices include restrictions on access to places of worship and participation in religious activities, participation in household chores, physical contact with male household members, and constraints on utilizing water sources [14, 19, 28, 33]. The western part of Nepal is known for the infamous *Chhaupadi*—the severest form of menstrual exclusion—that designates women as impure and untouchable during menstruation or postpartum period banishing them to separate makeshift houses referred to menstrual huts or cowshed [14, 26, 28, 34]. Despite legislative measures against menstrual discriminatory practices, recent studies reveal that the practice is still pervasive in the western part of the country [26, 35–38], subjecting women and girls to physical and mental health challenges, sexual abuse, snake bites and even death [28, 35, 39–42].

Only recently has the global health and social research community endeavored to confront menstrual hygiene management as a substantial development concern and a barrier to achieving gender equality [15, 18, 43–45]. The societal impact of menstrual hygiene mismanagement extends beyond adolescence, affecting adult women in their occupational engagements [4, 16, 46–48]. Recent studies have demonstrated a strong association between menstrual hygiene management and school performance or attendance among girls [30, 49, 50]. In addition, poor menstrual hygiene management has been linked to reproductive tract infections[51–53]. Unequal access to menstrual materials and lack of safe menstrual management spaces have been consistently identified as the key drivers of poor menstrual management across the globe [7, 8, 10–12, 16]. Whereas these findings coincide with the recent scale-up of WASH intervention programs [30, 48, 50, 54–56] providing hope to millions of girls and women in LMIC, further progress is likely if more research is dedicated to fighting the negative cultural beliefs and taboos surrounding menstrual hygiene management [2, 21]. Menstrual discriminatory practices vary across countries and geographical regions characterized by differences in sociodemographic characteristics that include religious and cultural diversity [25, 27, 28, 33, 40, 57]. Previous studies indicate regional distribution patterns across Nepal with particularly high prevalence estimates in Karnali and Sudurpashchim provinces [26, 29, 58]. However, these estimates mainly provide provincial-level estimates, excluding fine-scale variation that may exist within provinces. Therefore, the development of fine-scale risk stratification maps of

menstrual discriminatory practices means that the most vulnerable populations can be identified more accurately and prioritized with scarce resources. In addition, the risk maps can aid the establishment of spatial patterns of menstrual exclusions for contextualizing changes in the burden of menstrual stigma relative to interventional policies and programs.

Using the 2022 Nepal Demographic and Health Survey (NDHS) dataset, we explored the spatial patterns of *a priori*-defined menstrual restriction outcomes related to engagement in religious activities, performance of household activities/household interactions, and *Chhaupadi*. The second objective was to assess the socioeconomic and demographic factors associated with the geographical variations in the risk of menstrual discrimination. This is one of the few studies to describe the fine-scale spatial heterogeneity of menstrual discriminatory practices in Nepal and the first to do so among women aged 15–49 years using a nationally representative population-based survey.

## Materials and Methods

### Ethics

The 2022 NDHS protocol was developed and reviewed by the Nepal Health Research Council and the ICF Institutional Review Board. Written consent from the household heads was a prerequisite for conducting the interviews [59]. Here we conducted a secondary data analysis using de-identified data from the 2022 NDHS. This dataset is publicly available and does not require ethical approvals for the preparation of the manuscript. Permission to use the data was obtained from the DHS division at ICF International, following the completion of an online request form available on the DHS website [59].

### Study site

Nepal is a landlocked country situated in southern Asia between latitudes 26˚ and 30˚ N and longitudes 80˚ and 89˚ E. Nestled along the southern slopes of the Himalayas, Nepal shares borders with India to the east, west, and south, and China to the north. Covering an area of 147,516 square kilometers, the country has a population of approximately 29.2 million people, with females comprising 51.1%. The country is divided into seven provinces: Koshi, Madhesh, Lumbini, Karnali, Bagmati, Gandaki, and Sudurpashchim (**Fig 1**). These provinces are subdivided into 77 districts, housing 753 local-level municipalities, with 293 classified as urban and 460 as rural. Nepal has three ecological zones: Terai (plains), hills, and mountains. Approximately 53.7% of the population resides in the Terai region, with 40.2% in the hilly areas and only 6.1% in the mountainous regions. Approximately, 65% of the population lives in urban centers. The country boasts a diverse culture, comprising over 120 castes/ethnicities, with Hinduism being the predominant religion[60].

### Study design and data source

The 2022 NDHS dataset is a nationally representative population-based survey that is freely available upon request [59]. Details of the study design, consenting, data collection, supervision, and training procedures have been described elsewhere [61]. Briefly, the 2022 NDHS survey adopted a two-stage stratified sampling methodology that included the segmentation of each of the seven provinces into urban and rural categories, yielding a total of 14 sampling strata. Within each stratum, the sampling frame was organized based on administrative units, and probability-proportional-to-size selection was applied in the first stage. In the initial sampling phase, 476 primary sampling units (PSUs) were chosen, with the probability proportional to PSU size, and independent selection was done within each stratum. Out of the 476

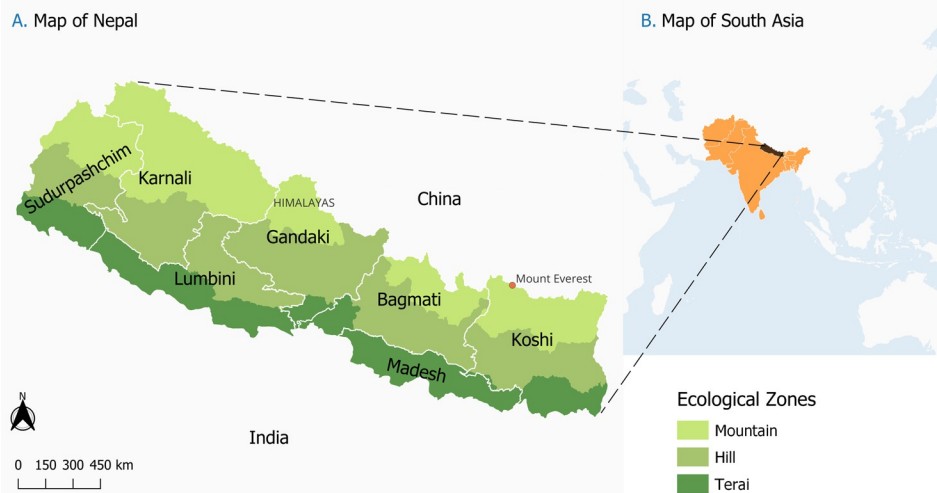

**Fig 1. Map of the study area showing the various ecological zones, first level administrative divisions and the neighboring countries.** *Base layer sources*: *South Asia shape file retrieved from the Natural Earth Map website (https:// www.naturalearthdata.com/)*, *under an open license (http://www.naturalearthdata.com/about/terms-of-use/)*; *shape files for Nepalese administrative boundaries were retrieved from (https://data.humdata.org/dataset/cod-ab-npl)*. *Maps were generated using open source QGIS version 3.34.2 (https://qgis.org/).*

PSUs, 248 were from urban areas, and 228 were from rural areas. Prior to the survey, household listing was conducted in all selected PSUs, establishing the sampling frame for the second-stage selection of sample households. Thirty households were chosen from each cluster leading to a total of 14,280 households. Global Positioning System (GPS) data were collected during the household listing exercise.

## Study participants

Our analyses are restricted to women and girls aged 15–49 years.

## Outcome variables

Our study sought to describe menstrual discriminatory practices among Nepalese women. These practices include restrictions on activities such as entering the temple, participating in religious activities, cooking, handling food items, fetching water, eating with the family, and the change in sleeping arrangements within the household during menstruation. In the 2022 NDHS, all eligible survey participants were asked whether or not were excluded from participating in these activities during menstruation (Yes/No). In our analyses, menstrual restriction activities were grouped into four binary composite outcomes: a) any form of menstrual restriction defined as any menstrual-related restriction, b) religious restrictions defined as exclusions from entering the temple or restrictions on participating in religious activities, c) household level restriction defined as restrictions from fetching water, cooking and handling food items, eating with the family, sleeping in the main house, and sleeping with the husband, and d) *Chhaupadi* defined as exclusion from sleeping in the main house as described previously [61]. For each composite binary outcome, {1} indicates an experience of at least one restriction within the defined sub-group and {0} otherwise.

## Predictor variables

Variations in menstrual restriction outcomes have been shown to vary at various geographical, socioeconomic, and demographic factors. Therefore, our potential predictor variables

included: participants' characteristics (age, ethnicity/caste, and education), household characteristics (age and sex of the household head, and wealth), and regional characteristics (urbanity and region (province) of residence). Detailed description of these variables and the construction of the derived variables has been reported previously [61] and summarized (**S1 Table**).

## Statistical analyses

The prevalence of menstrual restrictions by social demographic characteristics were estimated accounting for the study design. To account for the complex sampling design, the Taylor linearization approach was used to calculate the standard errors [62, 63].

## Spatial Analyses

The risk maps were created using an adaptive bandwidth in a weighted Gaussian kernel density estimation methodology [64] to yield spatially continuous risk maps and interpolate georeferenced outcomes of menstrual restriction. This will support the detection of high-risk areas for targeted interventions and resource allocation.

## Local cluster detection

We applied the Bernoulli probability model within SaTScan software to identify geographic areas with significantly higher menstrual restriction cases than expected by chance. We specified a circular, non-overlapping scanning window that systematically moves across the geographical space, with the radius varying from zero to a maximum radius determined based on a predetermined maximum enclosed population size (a priori specified to enclose up to 50% of the sample population in these analyses). For each location and window size, the software counts the observed cases whilst the expected cases are computed assuming a uniform distribution across the population. The scan statistic compares case counts inside and outside each circle, generating an observed log-likelihood statistic that is evaluated against the null hypothesis of complete spatial randomness. In this study, Monte-Carlo simulation with 999 replications was used to create permutations of the observed cases across all data locations generating the simulated log-likelihoods. The observed log-likelihood and the simulated log-likelihoods were used to calculate the p-value for statistical significance testing. The Gini coefficient determined optimal non-overlapping cluster sizes. Hereinafter, a cluster is defined as a geographical area experiencing significantly ($p < 0.05$) higher prevalence of menstrual discrimination than the average. No covariate adjustments were made in either the weighted Gaussian kernel density estimation or the local cluster analyses because our focus was to describe spatial variations in menstrual restrictions without accounting for specific causes.

## Regression analyses

We used a weighted modified Poisson regression model (weighted to accommodate the NDHS sampling design weights) to assess the association between menstrual restriction outcomes and socioeconomic and demographic factors. Modified Poisson regression with robust standard errors [65, 66] was used to assess the factors influencing heterogeneity in menstrual restriction outcomes. The variance inflation factors (VIF) were calculated to test for potential multicollinearity. To assess the extent to which variations on menstrual restrictions are accounted for by unmeasured variables at the regional level, we incorporated region (province) as a fixed effect in the regression model. McFadden's pseudo-$R2$ was used to quantify the contribution of each socioeconomic and demographic variable in the regression models. Statistical analyses were performed in software R version 4.3.2, local clusters were detected using

SaTScan version 10.1.2 and maps were produced using QGIS version 3.34.2. All analyses accounted for the survey design.

## Results

Of 14,845 women aged 15 to 49 years who participated in the 2022 NDHS, 1780 (12%) had incomplete data on menstruation. Therefore, our analysis included 13,065 (88%) study participants who reported at least one menstrual event within the last year preceding the survey. The distribution of the study participants by social-demographic characteristics are presented in (**S2 Table**)

### Prevalence of menstrual restrictions

The overall prevalence of experiencing any type of menstrual restriction in Nepal was 84.8% (95% CI: 84.2–85.4). Among the seven provinces, Sudurpashchim had the highest prevalence of experiencing any form of menstrual restrictions (95.6% [95% CI: 94.2–96.7]) followed by Karnali (91.7% [95%CI: 89.5–93.4]) whist Bagmati reported the lowest prevalence (79.0% [95% CI: 77.5–80.5]). Furthermore, the prevalence of experiencing any form of menstrual restriction was highest among older women (89.9% [95%CI: 87.7–91.7] among women aged 45–49), those with higher education (91.5% [95%CI: 89.0–93.4]), Brahmin/Chhetri ethnic background (96.1% [95%CI: 95.4–96.6]), residing in households led by men (85.6% [95%CI: 84.8–86.3]) or led by older household heads (88.0% [95%CI: 86.1–89.6] among household heads older than 64 years) and those residing in wealthy households (87.3% [95%CI: 86.0–88.5]). However, because of the high prevalence of experiencing any form of menstrual restrictions in this setting, variations within socioeconomic and demographic factors were moderate to small (**Table 1**). Among the other forms of menstrual discriminatory practices, religious restrictions were the most prevalent (79.8% [95%CI: 79.1–80.5]), followed by household-level restrictions (39.5% [95%CI: 38.7–40.4]) and *Chhaupadi* (6.2% [95%CI: 5.8–6.6]) in that order. Further description of the socioeconomic and demographic factors for each form of menstrual restriction is presented in **Table 1**.

### Spatial Heterogeneity of Menstrual Restrictions

At the region (province) level, there were moderate variations in the prevalence of any menstrual restriction (range: 79.0% to 95.6%) and religious restrictions (range: 73.8% to 85.2%). However, we observed marked regional variations in household level restrictions (range: 16.0% to 76.2%), and *Chhaupadi* (range: 1.8% to 26.1%). Geospatial analysis using the adaptive Gaussian kernel density methodology revealed clear spatial variation in the prevalence of menstrual restrictions across the study area. The kernel-derived prevalence of experiencing any type of menstrual restriction varied between 23.7% and 100%, with mild variation defined by the country's topography (**Fig 2A**). Similarly, the high prevalence of religious restrictions was geographically restricted to the densely populated *Terai* and Hilly areas (**Fig 2B**). Local cluster analyses of religious restrictions revealed 17 statistically significant high-prevalence clusters (p<0.05), however, the ratio between the risk inside and outside the clusters (RR range: 1.13 and 1.27) was small (**Table 2**) suggesting a near homogeneous distribution in prevalence across the country.

The kernel-derived prevalence of menstrual restriction within the household varied markedly over the geographical area, ranging between 0% and 100% and between 0% and 66.9% for *Chhaupadi*. These analyses reveal a clear region of high prevalence (middle and western region of Nepal) and low prevalence in the eastern region of Nepal (**Fig 2**). Local cluster analyses of household-level restrictions and *Chhaupadi* revealed 11 and 1 statistically

**Table 1. Prevalence of menstrual restrictions by social-demographic and economic factors.**

| Characteristic | Characteristic | Any Restrictions | | | Religious restrictions | | | Household level restrictions | | | *Chhaupadi* | | |
|---|---|---|---|---|---|---|---|---|---|---|---|---|---|
| | | N | Prevalence n (%) | (95% CI) | N | Prevalence n (%) | (95% CI) | N | Prevalence n (%) | (95% CI) | N | Prevalence n (%) | (95% CI) |
| Age | 15–19 | 2,520 | 2081 (82.6) | 81.0–84.0 | 2,520 | 1949 (77.3) | 75.7–78.9 | 2,520 | 1020 (40.5) | 38.6–42.4 | 2,520 | 186 (7.4) | 6.4–8.5 |
| | 20–24 | 2,338 | 1954 (83.6) | 82.0–85.0 | 2,338 | 1839 (78.7) | 76.9–80.3 | 2,337 | 849 (36.3) | 34.4–38.3 | 2,337 | 135 (5.8) | 4.9–6.8 |
| | 25–29 | 2,148 | 1799 (83.8) | 82.1–85.3 | 2,148 | 1691 (78.7) | 76.9–80.4 | 2,148 | 778 (36.2) | 34.2–38.3 | 2,148 | 105 (4.9) | 4.1–5.9 |
| | 30–34 | 1,937 | 1658 (85.6) | 84.0–87.1 | 1,937 | 1566 (80.8) | 79.0–82.5 | 1,937 | 764 (39.4) | 37.3–41.6 | 1,937 | 109 (5.6) | 4.7–6.7 |
| | 35–39 | 1,852 | 1584 (85.5) | 83.9–87.1 | 1,851 | 1504 (81.3) | 79.4–83.0 | 1,852 | 763 (41.2) | 39.0–43.5 | 1,852 | 105 (5.7) | 4.7–6.8 |
| | 40–44 | 1,389 | 1217 (87.6) | 85.8–89.2 | 1,389 | 1145 (82.4) | 80.3–84.3 | 1,388 | 573 (41.3) | 38.7–43.9 | 1,389 | 103 (7.4) | 6.2–8.9 |
| | 45–49 | 881 | 792 (89.9) | 87.7–91.7 | 882 | 734 (83.2) | 80.6–85.5 | 882 | 415 (47.1) | 43.8–50.4 | 882 | 63 (7.1) | 5.6–9.0 |
| Education | Basic | 4,084 | 3394 (83.1) | 81.9–84.2 | 4,084 | 3173 (77.7) | 76.4–78.9 | 4,083 | 1545 (37.8) | 36.4–39.3 | 4,084 | 246 (6) | 5.3–6.8 |
| | Higher | 623 | 570 (91.5) | 89.0–93.4 | 623 | 551 (88.4) | 85.7–90.7 | 623 | 329 (52.8) | 48.9–56.7 | 624 | 19 (3) | 2.0–4.7 |
| | No education | 3,036 | 2575 (84.8) | 83.5–86.0 | 3,037 | 2419 (79.7) | 78.2–81.0 | 3,037 | 976 (32.1) | 30.5–33.8 | 3,037 | 244 (8) | 7.1–9.1 |
| | Secondary | 5,322 | 4546 (85.4) | 84.4–86.3 | 5,322 | 4287 (80.6) | 79.5–81.6 | 5,321 | 2312 (43.5) | 42.1–44.8 | 5,321 | 297 (5.6) | 5.0–6.2 |
| Province | Bagmati | 2,763 | 2183 (79.0) | 77.5–80.5 | 2,763 | 2078 (75.2) | 73.6–76.8 | 2,762 | 1009 (36.5) | 34.8–38.3 | 2,763 | 51 (1.8) | 1.4–2.4 |
| | Gandaki | 1,292 | 1037 (80.3) | 78.0–82.3 | 1,292 | 953 (73.8) | 71.3–76.1 | 1,292 | 514 (39.8) | 37.1–42.5 | 1,292 | 46 (3.6) | 2.7–4.7 |
| | Karnali | 781 | 716 (91.7) | 89.5–93.4 | 782 | 650 (83.1) | 80.3–85.6 | 781 | 539 (69.0) | 65.7–72.2 | 782 | 204 (26.1) | 23.1–29.3 |
| | Koshi | 2,193 | 1895 (86.4) | 84.9–87.8 | 2,193 | 1825 (83.2) | 81.6–84.7 | 2,193 | 912 (41.6) | 39.5–43.7 | 2,193 | 72 (3.3) | 2.6–4.1 |
| | Lumbini | 2,457 | 2065 (84.0) | 82.5–85.4 | 2,456 | 1949 (79.4) | 77.7–80.9 | 2,457 | 985 (40.1) | 38.2–42.0 | 2,457 | 99 (4) | 3.3–4.9 |
| | Madhesh | 2,466 | 2126 (86.2) | 84.8–87.5 | 2,467 | 2101 (85.2) | 83.7–86.5 | 2,467 | 395 (16.0) | 14.6–17.5 | 2,466 | 116 (4.7) | 3.9–5.6 |
| | Sudurpashchim | 1,113 | 1064 (95.6) | 94.2–96.7 | 1,114 | 875 (78.5) | 76.0–80.9 | 1,113 | 808 (72.6) | 69.9–75.1 | 1,114 | 218 (19.6) | 17.3–22 |
| Ethnicity | Brahmin/Chhetri | 3,735 | 3588 (96.1) | 95.4–96.6 | 3,735 | 3253 (87.1) | 86.0–88.1 | 3,735 | 2847 (76.2) | 74.8–77.6 | 3,735 | 457 (12.2) | 11.2–13.3 |
| | Dalit | 1,933 | 1587 (82.1) | 80.3–83.7 | 1,934 | 1448 (74.9) | 72.9–76.8 | 1,934 | 761 (39.3) | 37.2–41.5 | 1,934 | 160 (8.3) | 7.1–9.6 |
| | Janajati | 4,879 | 3709 (76.0) | 74.8–77.2 | 4,878 | 3562 (73.0) | 71.8–74.2 | 4,879 | 974 (20.0) | 18.9–21.1 | 4,878 | 70 (1.4) | 1.1–1.8 |
| | Madhesi | 1,942 | 1778 (91.6) | 90.2–92.7 | 1,943 | 1756 (90.4) | 89.0–91.6 | 1,942 | 526 (27.1) | 25.2–29.1 | 1,942 | 117 (6) | 5.1–7.2 |
| | Muslim | 562 | 411 (73.1) | 69.3–76.6 | 562 | 399 (71.0) | 67.1–74.6 | 562 | 47 (8.4) | 6.3–10.9 | 562 | 0 (0) | 0–0.7 |
| | Other | 14 | 12 (85.7) | 60.1–96.0 | 14 | 12 (85.7) | 60.1–96.0 | 14 | 8 (57.1) | 32.6–78.6 | 14 | 2 (14.3) | 4.0–39.9 |
| Residence | Rural | 4,081 | 3391 (83.1) | 81.9–84.2 | 4,081 | 3126 (76.6) | 75.3–77.9 | 4,081 | 1672 (41.0) | 39.5–42.5 | 4,081 | 373 (9.1) | 8.3–10.1 |
| | Urban | 8,985 | 7695 (85.6) | 84.9–86.4 | 8,985 | 7304 (81.3) | 80.5–82.1 | 8,984 | 3491 (38.9) | 37.9–39.9 | 8,985 | 433 (4.8) | 4.4–5.3 |
| Gender of Household head | Female | 4,568 | 3811 (83.4) | 82.3–84.5 | 4,568 | 3593 (78.7) | 77.4–79.8 | 4,568 | 1647 (36.1) | 34.7–37.5 | 4,568 | 302 (6.6) | 5.9–7.4 |
| | Male | 8,497 | 7274 (85.6) | 84.8–86.3 | 8,497 | 6836 (80.5) | 79.6–81.3 | 8,497 | 3516 (41.4) | 40.3–42.4 | 8,497 | 504 (5.9) | 5.4–6.5 |
| Age of Household head | 15–24 | 615 | 492 (80.0) | 76.7–83.0 | 616 | 465 (75.5) | 71.9–78.7 | 616 | 167 (27.1) | 23.7–30.8 | 616 | 34 (5.5) | 4.0–7.6 |
| | 25–34 | 2,288 | 1898 (83.0) | 81.4–84.4 | 2,288 | 1795 (78.5) | 76.7–80.1 | 2,289 | 724 (31.6) | 29.8–33.6 | 2,288 | 124 (5.4) | 4.6–6.4 |
| | 35–44 | 3,938 | 3299 (83.8) | 82.6–84.9 | 3,937 | 3120 (79.2) | 78.0–80.5 | 3,938 | 1528 (38.8) | 37.3–40.3 | 3,938 | 256 (6.5) | 5.8–7.3 |
| | 45–54 | 3,161 | 2711 (85.8) | 84.5–86.9 | 3,161 | 2559 (81.0) | 79.5–82.3 | 3,161 | 1289 (40.8) | 39.1–42.5 | 3,161 | 192 (6.1) | 5.3–7.0 |
| | 55–64 | 1,773 | 1550 (87.4) | 85.8–88.9 | 1,773 | 1450 (81.8) | 79.9–83.5 | 1,773 | 797 (45.0) | 42.6–47.3 | 1,773 | 105 (5.9) | 4.9–7.1 |
| | Over 64 | 1,290 | 1135 (88.0) | 86.1–89.6 | 1,290 | 1039 (80.5) | 78.3–82.6 | 1,290 | 659 (51.1) | 48.4–53.8 | 1,290 | 95 (7.4) | 6.1–8.9 |
| Wealth index | Poorest | 2,280 | 1869 (82.0) | 80.3–83.5 | 2,281 | 1615 (70.8) | 68.9–72.6 | 2,281 | 1152 (50.5) | 48.5–52.6 | 2,281 | 331 (14.5) | 13.1–16.0 |
| | Poorer | 2,443 | 2077 (85.0) | 83.5–86.4 | 2,443 | 1953 (79.9) | 78.3–81.5 | 2,442 | 874 (35.8) | 33.9–37.7 | 2,443 | 155 (6.3) | 5.4–7.4 |
| | Middle | 2,636 | 2213 (84.0) | 82.5–85.3 | 2,636 | 2121 (80.5) | 78.9–81.9 | 2,636 | 882 (33.5) | 31.7–35.3 | 2,636 | 151 (5.7) | 4.9–6.7 |
| | Richer | 2,868 | 2449 (85.4) | 84.1–86.6 | 2,868 | 2343 (81.7) | 80.2–83.1 | 2,868 | 1004 (35) | 33.3–36.8 | 2,868 | 118 (4.1) | 3.4–4.9 |
| | Richest | 2,839 | 2478 (87.3) | 86.0–88.5 | 2,839 | 2398 (84.5) | 83.1–85.8 | 2,840 | 1251 (44) | 42.2–45.9 | 2,839 | 52 (1.8) | 1.4–2.4 |
| Total | | 13,065 | 11085 (84.8) | 84.2–85.4 | 13,065 | 10429 (79.8) | 79.1–80.5 | 13,065 | 5163 (39.5) | 38.7–40.4 | 13,065 | 806 (6.2) | 5.8–6.6 |

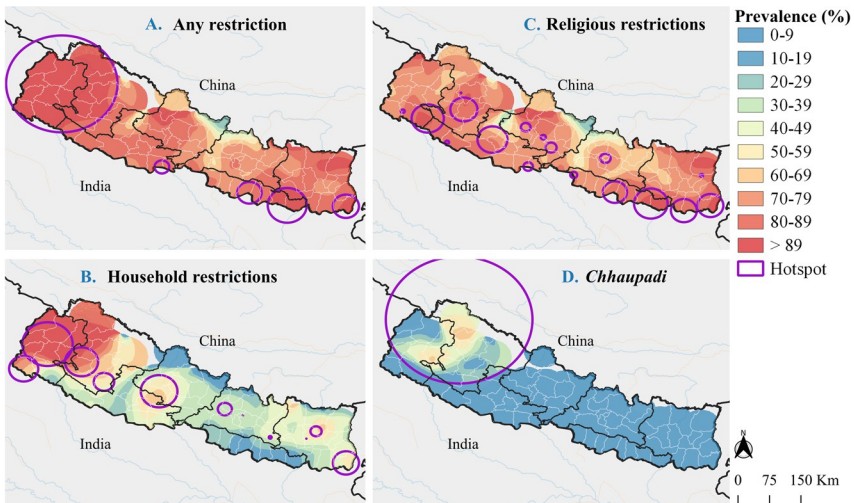

**Fig 2. Continuous surface maps of prevalence and clusters of high prevalence of menstrual discriminatory practices in Nepal.** Panels (**A**) to (**D**) show the spatial variations in the prevalence of the various forms of menstrual restrictions and the corresponding clusters of high prevalence. Shape files for Nepalese administrative boundaries were retrieved from (https://data.humdata.org/dataset/cod-ab-npl). Maps were generated using open source QGIS version 3.34.2 (https://qgis.org/).

significant clusters respectively. The prevalence ratios for the detected high-prevalence clusters ranged from 1.32 to 2.34 for household-level restrictions and 7.45 for *Chhaupadi*. In all the local cluster analyses, the primary clusters were located in either Sudurpashchim or Karnali, or both (**Fig 2** and **Table 2**).

## Predictors of menstrual restrictions

In the multivariable analysis, older women aged between 35–39, 40–44, and 45–49 were associated with a 5% (aPR = 1.05 [95%CI: 1.01–1.09], P = 0.021), 5% (aPR = 1.05 [95%CI: 1.01–1.10], P = 0.011), and 6% (aPR = 1.06 [95%CI: 1.01–1.11], P = 0.022) increase in the likelihood of experiencing restrictions on religious activities respectively. However, there was no clear association with household-level restrictions or *Chhaupadi*. Compared to women with no education, women with primary, secondary and higher education were associated with a 13% (aPR = 1.13 [95%CI: 1.05–1.22], P = 0.002), 10% (aPR = 1.10 [95%CI: 1.00–1.21], P = 0.041), and 17% (aPR = 1.17[95%CI: 1.03–1.34], P = 0.015) increase in the likelihood of experiencing household level restrictions, however, there was no evidence of associations between education and the other forms of menstrual restrictions. Ethnicity/caste was associated with variations in all forms of menstrual restrictions whilst rural/urban residence was only significantly associated with *Chhaupadi* in the univariable models but not in the multivariable models (**S3 Table**). Women residing in households led by a female head were associated with 11% (aPR = 0.89 [95%CI: 0.84–0.94], p<0.001) lower likelihood of experiencing household-level restrictions but no evidence of an association with restrictions on religious activities or *Chhaupadi* (**Table 3**). In addition, older household heads were associated with an increased risk of experiencing household-level menstrual restrictions and *Chhaupadi*. On one hand, increased household wealth was associated with an increased risk of experiencing restrictions on religious activities. On the other hand, increased household wealth was significantly associated with protection against *Chhaupadi* (**Table 3**).

**Table 2. Characteristics of clusters of high prevalence.**

| Restriction | Hotspot | Prevalence Risk | P-Value | Radius (KM) |
|---|---|---|---|---|
| Any restriction | 1 | 1.08 | 0.003 | 28.68 |
| | 2 | 1.11 | <0.001 | 30.79 |
| | 3 | 1.13 | <0.001 | 45.95 |
| | 4 | 1.13 | 0.013 | 17.23 |
| | 5 | 1.16 | <0.001 | 127.88 |
| Religious restrictions | 1 | 1.12 | 0.001 | 12.40 |
| | 2 | 1.13 | 0.02 | 35.18 |
| | 3 | 1.14 | <0.001 | 30.89 |
| | 4 | 1.16 | <0.001 | 28.68 |
| | 5 | 1.18 | <0.001 | 31.67 |
| | 6 | 1.18 | 0.014 | 6.07 |
| | 7 | 1.19 | <0.001 | 37.65 |
| | 8 | 1.20 | <0.001 | 30.79 |
| | 9 | 1.21 | 0.042 | 4.62 |
| | 10 | 1.22 | <0.001 | 40.03 |
| | 11 | 1.24 | <0.001 | 11.32 |
| | 12 | 1.24 | 0.003 | 13.28 |
| | 13 | 1.25 | 0.019 | 8.22 |
| | 14 | 1.27 | <0.001 | 0.89 |
| | 15 | 1.27 | 0.003 | 3.69 |
| | 16 | 1.27 | 0.005 | 11.26 |
| | 17 | 1.27 | 0.024 | 4.57 |
| Household restrictions | 1 | 1.32 | 0.001 | 30.79 |
| | 2 | 1.41 | 0.001 | 42.58 |
| | 3 | 1.57 | 0.001 | 33.67 |
| | 4 | 1.59 | 0.001 | 16.45 |
| | 5 | 1.69 | 0.001 | 23.68 |
| | 6 | 1.77 | 0.001 | 3.41 |
| | 7 | 1.83 | 0.001 | 13.14 |
| | 8 | 1.86 | 0.024 | 0.00 |
| | 9 | 1.91 | 0.001 | 38.12 |
| | 10 | 2.14 | 0.001 | 0.00 |
| | 11 | 2.34 | 0.001 | 57.79 |
| *Chhaupadi* | 1 | 7.45 | <0.001 | 169.05 |

The final multivariable regression model accounted for 4%, 3%, 24%, and 23% of the variability in any form of menstrual restriction, restriction on religious activities, household level restrictions, and *Chhaupadi* respectively. However, the variability accounted for by the variable ethnicity/caste was 3% for any form of menstrual restriction, 2% for restriction on religious activities, 19% for household level restrictions and 11% for *Chhaupadi*.

## Discussion

In the global pursuit of improved Water, Sanitation, and Hygiene (WASH), menstrual health and hygiene (MHH) persist as pressing challenges in low and middle-income countries (LMICs) directly contributing to socio-economic inequity. Although there has been significant progress in other areas of WASH, millions of women still encounter obstacles in managing

**Table 3. Multivariable Poisson regression of menstrual restrictions by social-demographic and economic factors.**

| Characteristic | Model1: Any restriction | | | Model2: Religious restrictions | | | Model3: Household level restrictions | | | Model4: *Chhaupadi* | | |
|---|---|---|---|---|---|---|---|---|---|---|---|---|
| | aPR | 95% CI | P-value | aPR | 95% CI | P-value | aPR | 95% CI | P-value | aPR | 95% CI | P-value |
| **Age (ref: 15–19 years)** | | | | | | | | | | | | |
| 20–24 | 1.01 | 0.98–1.03 | 0.723 | 1.00 | 0.97–1.04 | 0.852 | 0.94 | 0.89–1.01 | 0.090 | 0.93 | 0.76–1.13 | 0.450 |
| 25–29 | 1.01 | 0.98–1.04 | 0.601 | 1.00 | 0.97–1.04 | 0.830 | 0.95 | 0.88–1.03 | 0.246 | 0.83 | 0.65–1.07 | 0.154 |
| 30–34 | 1.03 | 1.00–1.07 | 0.064 | 1.03 | 0.99–1.07 | 0.115 | 1.03 | 0.95–1.12 | 0.463 | 0.93 | 0.73–1.19 | 0.567 |
| 35–39 | 1.04 | 1.01–1.08 | 0.024 | 1.05 | 1.01–1.09 | 0.021 | 1.03 | 0.95–1.12 | 0.483 | 0.87 | 0.67–1.12 | 0.285 |
| 40–44 | 1.06 | 1.02–1.10 | 0.001 | 1.05 | 1.01–1.10 | 0.011 | 1.02 | 0.94–1.11 | 0.643 | 1.12 | 0.82–1.53 | 0.484 |
| 45–49 | 1.07 | 1.03–1.12 | 0.001 | 1.06 | 1.01–1.11 | 0.022 | 1.09 | 0.99–1.21 | 0.070 | 1.05 | 0.76–1.45 | 0.754 |
| **Highest level of education (ref: No education)** | | | | | | | | | | | | |
| Basic | 0.99 | 0.96–1.02 | 0.599 | 0.98 | 0.95–1.01 | 0.220 | 1.13 | 1.05–1.22 | 0.002 | 0.94 | 0.76–1.17 | 0.600 |
| Secondary | 0.99 | 0.96–1.02 | 0.642 | 0.98 | 0.95–1.02 | 0.339 | 1.10 | 1.00–1.21 | 0.041 | 0.92 | 0.72–1.18 | 0.529 |
| Higher | 1.03 | 0.98–1.08 | 0.224 | 1.03 | 0.98–1.09 | 0.192 | 1.17 | 1.03–1.34 | 0.015 | 0.95 | 0.55–1.65 | 0.863 |
| **Residence (ref: Rural)** | | | | | | | | | | | | |
| Urban | 1.02 | 0.98–1.05 | 0.37 | 1.02 | 0.98–1.06 | 0.378 | 1.02 | 0.93–1.11 | 0.707 | 0.81 | 0.63–1.05 | 0.11 |
| **Caste/Ethnicity (ref: Janajati)** | | | | | | | | | | | | |
| Brahmin/Chhetri | 1.24 | 1.20–1.28 | <0.001 | 1.19 | 1.14–1.23 | <0.001 | 3.53 | 3.15–3.95 | <0.001 | 5.41 | 3.76–7.78 | <0.001 |
| Dalit | 1.07 | 1.03–1.12 | 0.002 | 1.02 | 0.97–1.08 | 0.376 | 2.26 | 1.97–2.59 | <0.001 | 4.00 | 2.70–5.92 | <0.001 |
| Madhesi | 1.17 | 1.12–1.22 | <0.001 | 1.15 | 1.09–1.20 | <0.001 | 2.08 | 1.68–2.58 | <0.001 | 4.75 | 2.61–8.63 | <0.001 |
| Muslim | 0.93 | 0.81–1.07 | 0.299 | 0.90 | 0.78–1.04 | 0.157 | 0.64 | 0.42–0.99 | 0.047 | 0.00 | 0.00–0.00 | <0.001 |
| Other | 1.12 | 0.85–1.48 | 0.420 | 1.13 | 0.85–1.49 | 0.397 | 3.06 | 1.62–5.75 | 0.001 | 13.57 | 1.98–92.98 | 0.008 |
| **Region/Province (ref: Bagmati)** | | | | | | | | | | | | |
| Gandaki | 1.04 | 0.96–1.12 | 0.325 | 1.01 | 0.93–1.11 | 0.791 | 1.13 | 0.98–1.32 | 0.103 | 1.57 | 0.93–2.64 | 0.094 |
| Karnali | 1.14 | 1.08–1.22 | <0.001 | 1.17 | 1.09–1.25 | <0.001 | 1.36 | 1.22–1.53 | <0.001 | 5.19 | 3.29–8.18 | <0.001 |
| Koshi | 1.13 | 1.06–1.20 | <0.001 | 1.16 | 1.09–1.24 | <0.001 | 1.22 | 1.07–1.40 | 0.004 | 1.34 | 0.75–2.39 | 0.329 |
| Lumbini | 1.08 | 1.02–1.14 | 0.014 | 1.08 | 1.02–1.16 | 0.016 | 1.06 | 0.93–1.22 | 0.370 | 1.46 | 0.81–2.62 | 0.206 |
| Madhesh | 1.11 | 1.03–1.19 | 0.005 | 1.16 | 1.08–1.26 | <0.001 | 0.47 | 0.36–0.61 | <0.001 | 1.40 | 0.72–2.70 | 0.319 |
| Sudurpashchim | 1.19 | 1.13–1.26 | <0.001 | 1.07 | 1.00–1.16 | 0.063 | 1.54 | 1.38–1.72 | <0.001 | 4.70 | 2.92–7.57 | <0.001 |
| **Gender of the household head (ref: Male)** | | | | | | | | | | | | |
| Female | 0.99 | 0.97–1.01 | 0.274 | 1.00 | 0.97–1.02 | 0.792 | 0.89 | 0.84–0.94 | <0.001 | 1.04 | 0.89–1.2 | 0.646 |
| **Age of the household head (ref 15–24 years)** | | | | | | | | | | | | |
| 25–34 | 1.03 | 0.97–1.09 | 0.412 | 1.02 | 0.96–1.09 | 0.545 | 1.24 | 1.06–1.46 | 0.008 | 1.19 | 0.85–1.67 | 0.310 |
| 35–44 | 1.02 | 0.96–1.08 | 0.603 | 1.00 | 0.94–1.07 | 0.919 | 1.55 | 1.34–1.79 | <0.001 | 1.47 | 1.05–2.06 | 0.024 |
| 45–54 | 1.04 | 0.98–1.10 | 0.191 | 1.03 | 0.96–1.10 | 0.394 | 1.62 | 1.39–1.89 | <0.001 | 1.52 | 1.08–2.14 | 0.018 |
| 55–64 | 1.06 | 1.00–1.13 | 0.046 | 1.04 | 0.97–1.11 | 0.266 | 1.78 | 1.52–2.07 | <0.001 | 1.48 | 1.05–2.08 | 0.025 |
| Over 64 | 1.06 | 1.00–1.13 | 0.049 | 1.02 | 0.95–1.09 | 0.659 | 1.97 | 1.69–2.29 | <0.001 | 1.89 | 1.31–2.74 | 0.001 |
| **Wealth Index (ref: Poorest)** | | | | | | | | | | | | |
| Poorer | 1.07 | 1.03–1.12 | 0.001 | 1.15 | 1.09–1.21 | <0.001 | 0.96 | 0.89–1.04 | 0.297 | 0.79 | 0.62–1.01 | 0.058 |
| Middle | 1.06 | 1.01–1.12 | 0.012 | 1.15 | 1.09–1.22 | <0.001 | 0.92 | 0.83–1.02 | 0.110 | 0.81 | 0.62–1.07 | 0.144 |
| Richer | 1.09 | 1.03–1.14 | 0.001 | 1.18 | 1.11–1.25 | <0.001 | 0.91 | 0.82–1.01 | 0.078 | 0.58 | 0.43–0.78 | <0.001 |
| Richest | 1.11 | 1.05–1.18 | <0.001 | 1.23 | 1.16–1.32 | <0.001 | 0.90 | 0.80–1.02 | 0.091 | 0.26 | 0.17–0.39 | <0.001 |

their menstrual periods safely and with dignity (Crawford et al., 2014; Health, 2018; Health–Americas, 2022). Our results indicate that Nepal continues to bear a disproportionate burden of menstrual-related discrimination, which manifests as exclusions from participating in social and religious activities. These findings are consistent with recent studies showing that *Chhaupadi* is most prevalent among the Brahmin/Chhetri ethnic community in the mid-western

part of Nepal [13, 28, 29, 34, 37]. Menstrual impurity is a concept rooted in Hinduism, with religious leaders often upholding this practice [34]. The Brahmins, who frequently serve as priests, along with high-caste Hindu women like Chhetris, are often expected to observe and enforce these rituals [67].

Overall, 84.8% of women in Nepal experienced some form of menstrual restriction that was characterized by mild geographical variations, likely due to the high prevalence of menstrual restrictions observed across the country. Among the forms of menstrual discriminatory practices considered, restrictions on religious activities were the most prevalent (79.8%). However, this reflects a 13% decline from the 2019 estimates [29], potentially resulting from the sustained mitigation measures from government and non-governmental agencies against menstrual discriminatory practices [14, 68]. The prevalence of *Chhaupadi* was highest in the mid and western part of Nepal, highlighting the role of regional, ethnic, and religious dynamics in shaping menstrual discriminatory practices. Our estimates in Karnali and Sudurpashchim provinces were similar to the 2020 estimates [29], reflecting minimal gains in the efforts to eradicate harmful practices. Achham and Deilekh districts were associated with the highest *Chhaupadi* case count per square kilometre (**Fig 3B**). This is noteworthy, considering that the two districts have previously been targeted with interventions that included the destruction of the menstrual huts and prosecution of those advocating/practicing the vice [35, 38, 69]. Our findings are consistent with prior studies that cite gaps in the enforcement of legislative interventions aimed at curbing the practice [36, 38, 69].

Local cluster analyses identified specific areas in Sudurpashchim and Karnali as high-prevalence areas for all forms of menstrual restrictions. The *Chhaupadi* prevalence clusters (**Fig 2**A) overlapped with clusters of absolute case counts (**Fig 3**B), suggesting that targeting these high-prevalence areas will likely lead to a significant overall decline in *Chhaupadi* case counts across the country. However, this does not hold for the other types of menstrual restrictions, in which the case counts follow the population density distribution, with notable clusters observed in the urban setting of Kathmandu where the prevalence is relatively low (**Fig 2**A compared to **Fig 3**A).

An important finding of our study is the continued practice of *Chhaupadi* despite legislative interventions thus highlighting the intricate interplay between cultural norms and legal frameworks [70]. This highlights the need to go beyond legislation and emphasize culturally sensitive community engagement and active involvement of local and religious leaders to mitigate the deeply rooted harmful cultural norms [36, 56]. The risk maps generated provide a visual representation of the geographical distribution of the various menstrual discriminatory practices. This spatial intelligence tool can be used by policymakers and health practitioners to strategically allocate resources by tailoring interventions to the needs of each region. The recognition of the high-prevalence clusters as focal points allow for the development of interventions characterized by cultural sensitivity. These interventions may include targeted education programs and sensitization to challenge and transform the deeply rooted beliefs perpetuating menstrual stigma [20].

Untargeted interventions, while well-intentioned, may fall short in areas where menstrual discriminatory practices are deeply ingrained. We contend that high-prevalence clusters, if left unaddressed, can perpetuate the harmful cultural norms thus hindering the broader efforts to eliminate the vice across the country. We hypothesize that directing community-based interventions towards these specific clusters could be a highly effective way to break the cycle of harmful cultural beliefs and expedite progress toward a society where women can manage their menstrual health without facing discrimination. In doing so, we advocate for an approach that not only aims to eliminate discriminatory practices but also seeks to empower women, challenge harmful cultural beliefs, and promote an environment of inclusivity.

Women residing in households led by older household heads were associated with a higher likelihood of experiencing *Chhaupadi* or household-level restrictions. This observation suggests

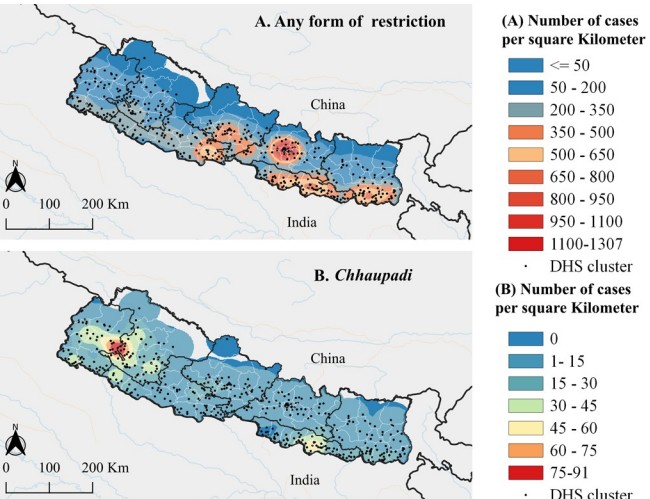

**Fig 3.** Continuous surface map showing the spatial case counts of individuals who experienced any form of menstrual restriction (panel **A**) and *Chhaupadi* (panel **B**), and the locations of demographic and health survey clusters. Shape files for Nepalese administrative boundaries were retrieved from (https://data.humdata.org/dataset/cod-ab-npl). Maps were generated using open source QGIS version 3.34.2 (https://qgis.org/).

a potential manifestation of conservative attitudes towards culture within the elderly demographic. In addition, women residing in households led by female household heads were associated with lower likelihoods of experiencing *Chhaupadi* or household-level restrictions, suggesting a potential role of female leadership in challenging cultural norms. In keeping with previous findings, ethnicity/caste, and wealth of the household emerged as key determinants of menstrual restrictions [57, 70]. Whilst richer households were associated with an increased likelihood of experiencing restrictions on religious activities, women residing in richer households were protected against *Chhaupadi*. Furthermore, older women were associated with an increased likelihood of experiencing restrictions on religious activities, whilst education was associated with a heightened risk of menstrual restrictions within the household, but was not associated with *Chhaupadi*. Ethnicity/caste was associated with variations in all forms of menstrual-restrictions, indicating the intersectionality of social-demographic and cultural factors [20].

Despite the robust statistical approaches used in this study, a substantial portion of the variability in the regression models remained unexplained. Therefore, there is a need for additional research to identify the unmeasured determinants of menstrual discrimination that may act at various geographic scales. Understanding these unmeasured predictor variables may enhance the development of interventions that address the root cause of menstrual discrimination. Interventions designed against menstrual discriminatory practices should not only target high-prevalence clusters but also incorporate a research component to test their utility and cost-effectiveness in real-world settings.

The significance of our study lies not only in its national scope but also in its novel approach to understanding the geographical nuances of menstrual health challenges in Nepal. Our study is the first to explore the micro-geographical variation of menstrual discriminatory practices against women 15–49 years of age. By delving into the geographical nuances of menstrual restrictions related to religious activities, household level, and *Chhaupadi*, we aim to guide targeted interventions and resource allocation.

Although our analyses provide a detailed description of menstrual discriminatory practices against girls and women in Nepal, we recognize some limitations. First, we could not establish

whether, during *Chhaupadi*, menstruating girls and women slept in makeshift huts, cowsheds, separate houses within the homestead, or separate rooms within the main house. Future studies may consider the various sleeping places during *Chhaupadi* to understand the evolving dynamics of the practice. Another limitation of our study is that we did not distinguish between internalized and externalized stigma to assess whether the practices were forced or voluntary. In addition, our analyses relied on reported cross-sectional data that may be subject to social desirability and recall bias and cannot establish causality. However, our regional estimates of prevalences are consistent with previous qualitative and quantitative studies[26, 28, 29] suggesting that the bias is likely minimal. Lastly, our study was conducted in a country experiencing high prevalences of menstrual discrimination resulting from deeply rooted religious and cultural norms. Therefore, our findings can only be generalized to regions with similar religious and cultural contexts and may not apply to other settings.

In conclusion, our study provides valuable insights into the complexity of menstrual health challenges in Nepal, shedding light on the spatial patterns and determinants of menstrual discriminatory practices. By elucidating the geographical variations, our findings provide a foundation for targeted community-based interventions such as cultural behaviour change education [20, 34, 45]. The identification and prioritization of vulnerable populations are important steps toward dismantling the deeply rooted beliefs and promoting a more inclusive and equitable society. As Nepal progresses towards sustainable development, addressing menstrual health disparities is imperative for achieving gender equality and societal well-being. Our study calls for sustained efforts, tailored community-based interventions, community engagement, and policy initiatives to alleviate menstrual stigma and discrimination against women and girls. Given that Chhaupadi is deeply ingrained in complex cultural and religious contexts, our study recommends strategies targeting key enforcers, such as religious leaders, mothers, heads, and older family members, who play critical roles in perpetuating menstrual taboos and rituals. These policy interventions may include a combination of women empowerment, advocacy, and educational awareness of the current laws illegalizing menstrual discrimination, and associated health risks [22, 28, 34].

## Supporting information

**S1 Table. Factors associated with menstrual taboos/restrictions.**
(DOCX)

**S2 Table. Social-demographic and economic characteristics of survey participants.**
(DOCX)

**S3 Table. Univariable Poisson regression of menstrual restrictions by social-demographic and economic factors.**
(DOCX)

**S4 Table. Model goodness-of-fit.**
(DOCX)

## Author Contributions

**Conceptualization:** Geoffrey Barini, Polycarp Mogeni.

**Data curation:** Geoffrey Barini, Polycarp Mogeni.

**Formal analysis:** Geoffrey Barini, Polycarp Mogeni.

**Investigation:** Geoffrey Barini, Sharon Amima, Damaris Mulwa, Polycarp Mogeni.

**Methodology:** Geoffrey Barini, Sharon Amima, Damaris Mulwa, Polycarp Mogeni.

**Project administration:** Polycarp Mogeni.

**Resources:** Polycarp Mogeni.

**Software:** Geoffrey Barini.

**Supervision:** Geoffrey Barini, Polycarp Mogeni.

**Validation:** Geoffrey Barini, Polycarp Mogeni.

**Visualization:** Geoffrey Barini.

**Writing – original draft:** Geoffrey Barini, Sharon Amima, Damaris Mulwa, Polycarp Mogeni.

**Writing – review & editing:** Geoffrey Barini, Sharon Amima, Damaris Mulwa, Polycarp Mogeni.

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
