## [Decision Letter · Decision Letter 0]

7 Aug 2024

PGPH-D-24-00629

Spatial heterogeneity of menstrual discriminatory practices against Nepalese women: a population-based study using the 2022 Demographic and Health Survey

Dear Dr. BARINI,

Thank you for submitting your manuscript to PLOS Global Public Health. After careful consideration, we feel that it has merit but does not fully meet PLOS Global Public Health’s publication criteria as it currently stands. Therefore, we invite you to submit a revised version of the manuscript that addresses the points raised during the review process.

 Your paper has huge potential to the field provided it addresses the concerns raised by our reviewers. Our first reviewer has given excellent feedback and I encourage the authors to address the feedback and revise the paper.

We look forward to receive the revised version at the earliest.

We regret for the delay in getting the review complete for the paper. We faced enormous amount of difficulty in finding right reviewers for the paper. I hope you understand our challenges!

We look forward to receiving your revised manuscript.

Kind regards,

Muthusamy Sivakami

Academic Editor

Journal Requirements:

Additional Editor Comments (if provided):

Thank you for your paper. Your paper has huge potential to the field provided it addresses the concerns raised by our reviewers. Our first reviewer has given excellent feedback and I encourage the authors to address the feedback and revise the paper.

We look forward to receive the revised version at the earliest.

We regret for the delay in getting the review complete for the paper. We faced enormous amount of difficulty in finding right reviewers for the paper. I hope you understand our challenges!

Reviewers' comments:

Reviewer's Responses to Questions

**Comments to the Author**

1. Does this manuscript meet PLOS Global Public Health’s publication criteria? Is the manuscript technically sound, and do the data support the conclusions? The manuscript must describe methodologically and ethically rigorous research with conclusions that are appropriately drawn based on the data presented.

Reviewer #1: Yes

Reviewer #2: Yes

2. Has the statistical analysis been performed appropriately and rigorously?

Reviewer #1: Yes

Reviewer #2: Yes

3. Have the authors made all data underlying the findings in their manuscript fully available (please refer to the Data Availability Statement at the start of the manuscript PDF file)?

Reviewer #1: Yes

Reviewer #2: No

4. Is the manuscript presented in an intelligible fashion and written in standard English?

Reviewer #1: Yes

Reviewer #2: Yes

5. Review Comments to the Author

Reviewer #1: I appreciate the opportunity to review this manuscript, which explores the spatial heterogeneity of menstrual discriminatory practices against Nepalese women using DHS survey data from 2022. I find the manuscript promising and worthy of publication. Below are some suggestions for improvement:

Given that all four authors are from Kenya and are working with data from Nepal, it would be beneficial for the authors to provide justification for the use of Nepalese data, demonstrating their understanding of the background and context.

Abstract:

Clarify the sample size in the abstract.

In the results section of the abstract, include corresponding beta coefficients and p-values for clarity.

Paper:

The literature around menstrual health in Nepal appears limited, as evidenced by the reliance on global literature in the introduction. Consider citing literature from neighboring countries like India and Bangladesh, which share similar socio-economic contexts.

While the abstract mentions that menstrual restrictions affect the SDGs, this discussion is not reflected in the introduction. Consider integrating this aspect into the introduction.

Motivate the importance of menstrual health by framing it as a public health and human rights issue, drawing on literature such as Babbar et al. (2022) and Hennegan (2017).

Strengthen the literature review to justify the selection of covariates, incorporating additional relevant literature, such as Anand et al. (2015), Almeida-Velasco & Sivakami (2019), Babbar et al. (2021), Babbar & Garikipati (2023), Singh et al. (2023), Chakrabarty et al. (2024), Goli et al. (2020), and Roy et al. (2020).

Provide a table explaining how control variables are created for clarity.

Explain the Taylor linearization approach used to account for the complex sampling design and clustering.

Clarify the rationale behind creating risk maps and conducting poison regression. It's essential to explain to readers why these methods are chosen and how they contribute to the analysis, rather than simply stating their use.

References

Anand, E., Singh, J. and Unisa, S. (2015) ‘Menstrual hygiene practices and its association with reproductive tract infections and abnormal vaginal discharge among women in India’, Sexual and Reproductive Healthcare 6(4): 249–254. https://doi.org/10.1016/j.srhc.2015.06.001

Almeida-Velasco, A. and Sivakami, M. (2019) ‘Menstrual hygiene management and reproductive tract infections: a comparison between rural and urban India’, Waterlines 38(2): 94–112. https://doi.org/10.3362/1756-3488.18-00032.

Babbar, K., Saluja, D., & Sivakami, M. (2021) 'How socio-demographic and mass media factors affect sanitary item usage among women in rural and urban India', Waterlines 40(3): 160-178. https://doi.org/10.3362/1756-3488.21-00003

Babbar, K., & Garikipati, S. (2023). What socio-demographic factors support disposable vs. sustainable menstrual choices? Evidence from India’s National Family Health Survey-5. Plos one, 18(8), e0290350.

Chakrabarty, M., Singh, A., Let, S., & Singh, S. (2023). Decomposing the rural–urban gap in hygienic material use during menstruation among adolescent women in India. Scientific Reports, 13(1), 22427.

Singh, A., & Chakrabarty, M. (2023). Spatial heterogeneity in the exclusive use of hygienic materials during menstruation among women in urban India. PeerJ, 11, e15026.

Goli, S., Sharif, N., Paul, S. and Salve, P.S. (2020) ‘Geographical disparity and socio-demographic correlates of menstrual absorbent use in India: a cross-sectional study of girls aged 15–24 years’, Children and Youth Services Review 117: 105283 https://doi.org/10.1016/j.childyouth.2020.105283.

Roy, A., Paul, P., Saha, J., Barman, B., Kapasia, N. and Chouhan, P. (2020) ‘Prevalence and correlates of menstrual hygiene practices among young currently married women aged 15–24 years: an analysis from a nationally representative survey of India’, European Journal of Contraception and Reproductive Health Care 26(1): 1–10. https://doi.org/10.1080/13625187.2020.1810227.

Reviewer #2: Title: Geographical Disparities in Menstrual Restrictions among Nepalese Women: A Critical Peer Review

Overall, the study "Geographical Disparities in Menstrual Restrictions among Nepalese Women" provides valuable insights into the prevalence and spatial distribution of menstrual discriminatory practices in Nepal. The research addresses an important issue related to gender equality, health, education, and sanitation, which aligns with the Sustainable Development Goals. However, there are several areas that require further clarification and improvement.

1. Methodology:

a. Data Collection: Information on how menstrual restrictions were measured and assessed within the survey should be provided. This would help readers understand the reliability and validity of the data used for analysis.

b. Statistical Analysis: While the study employed appropriate statistical methods such as Poisson regression and spatial scan statistics, additional information on model assumptions, goodness-of-fit tests, and potential confounders should be included to strengthen the analysis.

2. Results:

a. Interpretation of Findings: The authors present prevalence rates for different forms of menstrual restrictions across regions but do not discuss potential reasons for these variations or compare them with previous studies. Providing a contextual interpretation of these findings would enhance their significance.

b. Spatial Analysis: The study utilizes kernel density estimation and local cluster detection techniques to identify high-prevalence clusters of menstrual restrictions. However, it is unclear how these clusters were defined or what criteria were used to determine their significance.

3. Discussion:

a. Limitations: The authors acknowledge limitations related to self-reported data and social desirability bias but do not discuss other potential sources of bias or limitations inherent in cross-sectional surveys.

b. Generalizability: While the study focuses on Nepal, it does not discuss whether its findings can be extrapolated to other countries or regions facing similar challenges with menstrual discrimination.

4. Implications:

a. Policy Recommendations: The study briefly mentions targeted interventions but does not provide specific recommendations for policymakers or practitioners working towards eliminating menstrual discriminatory practices in Nepal.

b. Future Research Directions: It would be beneficial if the authors suggest areas for future research based on their findings, such as exploring cultural norms surrounding menstruation or evaluating existing interventions aimed at reducing menstrual restrictions.

Typos in the manuscript e.g on line 310: change “We hypothesis” to “We hypothesize”

In summary , this study provides valuable insights into the spatial heterogeneity of menstrual discriminatory practices among Nepalese women using robust statistical methods on population-based data from a reputable survey source.

6. PLOS authors have the option to publish the peer review history of their article (what does this mean?). If published, this will include your full peer review and any attached files.

**Do you want your identity to be public for this peer review?** For information about this choice, including consent withdrawal, please see our Privacy Policy.

Reviewer #1: No

Reviewer #2: No

---

## [Decision Letter · Decision Letter 1]

17 Sep 2024

Spatial heterogeneity of menstrual discriminatory practices against Nepalese women: a population-based study using the 2022 Demographic and Health Survey

PGPH-D-24-00629R1

Dear DR. BARINI,

We are pleased to inform you that your manuscript 'Spatial heterogeneity of menstrual discriminatory practices against Nepalese women: a population-based study using the 2022 Demographic and Health Survey' has been provisionally accepted for publication in PLOS Global Public Health.

Best regards,

Muthusamy Sivakami

Academic Editor

Thank you for carrying out the revisions to the satisfaction of the original reviewer. We are happy to accept the paper provided the necessary editorial changes are made when asked.

Reviewer Comments (if any, and for reference):

Reviewer's Responses to Questions

**Comments to the Author**

1. If the authors have adequately addressed your comments raised in a previous round of review and you feel that this manuscript is now acceptable for publication, you may indicate that here to bypass the “Comments to the Author” section, enter your conflict of interest statement in the “Confidential to Editor” section, and submit your "Accept" recommendation.

Reviewer #1: (No Response)

2. Does this manuscript meet PLOS Global Public Health’s publication criteria? Is the manuscript technically sound, and do the data support the conclusions? The manuscript must describe methodologically and ethically rigorous research with conclusions that are appropriately drawn based on the data presented.

Reviewer #1: (No Response)

3. Has the statistical analysis been performed appropriately and rigorously?

Reviewer #1: (No Response)

4. Have the authors made all data underlying the findings in their manuscript fully available (please refer to the Data Availability Statement at the start of the manuscript PDF file)?

Reviewer #1: (No Response)

5. Is the manuscript presented in an intelligible fashion and written in standard English?

Reviewer #1: (No Response)

6. Review Comments to the Author

Reviewer #1: (No Response)

7. PLOS authors have the option to publish the peer review history of their article (what does this mean?). If published, this will include your full peer review and any attached files.

**Do you want your identity to be public for this peer review?** For information about this choice, including consent withdrawal, please see our Privacy Policy.

Reviewer #1: No
